# Highly Efficient Production of Cellulosic Ethanol from Poplar Using an Optimal C6/C5 Co-Fermentation Strain of *Saccharomyces cerevisiae*

**DOI:** 10.3390/microorganisms12061174

**Published:** 2024-06-09

**Authors:** Fadi Xu, Dongming Sun, Zhaojiang Wang, Menglei Li, Xiaolong Yin, Hongxing Li, Lili Xu, Jianzhi Zhao, Xiaoming Bao

**Affiliations:** Key Laboratory of Biobased Material and Green Papermaking, School of Bioengineering, Qilu University of Technology, Shandong Academy of Sciences, 3501 Daxue Road, Jinan 250353, China

**Keywords:** poplar, pretreatment, enzymolysis, cellulosic ethanol, C5/C6 co-fermentation *Saccharomyces cerevisiae*

## Abstract

Cellulosic ethanol is the key technology to alleviate the pressure of energy supply and climate change. However, the ethanol production process, which is close to industrial production and has a high saccharification rate and ethanol yield, still needs to be developed. This study demonstrates the effective conversion of poplar wood waste into fuel-grade ethanol. By employing a two-step pretreatment using sodium chlorite (SC)-dilute sulfuric acid (DSA), the raw material achieved a sugar conversion rate exceeding 85% of the theoretical value. Under optimized conditions, brewing yeast co-utilizing C6/C5 enabled a yield of 35 g/L ethanol from 10% solid loading delignified poplar hydrolysate. We increased the solid loading to enhance the final ethanol concentration and optimized both the hydrolysis and fermentation stages. With 20% solid loading delignified poplar hydrolysate, the final ethanol concentration reached 60 g/L, a 71.4% increase from the 10% solid loading. Our work incorporates the pretreatment, enzymatic hydrolysis, and fermentation stages to establish a simple, crude poplar waste fuel ethanol process, expanding the range of feedstocks for second-generation fuel ethanol production.

## 1. Introduction

With the challenges posed by global energy demand, environmental pollution, and climate change, the development of environmentally friendly and renewable energy sources has become a necessary trend [1]. Europe, the Americas, and other major economic regions are actively pursuing alternative materials and technologies to diminish their reliance on fossil fuels. Their objective is to progressively supplant fossil fuels with renewable and sustainable alternatives. Therefore, there is great interest in using plant biomass for conversion to clean fuels [2]. Biofuels stand as the foremost liquid renewable fuel. Notably, biofuels boast compatibility with fossil fuels, it can be used in vehicles when mixed in proportion [3]. Biofuel ethanol is an ideal gasoline filler that has received global attention as a transportation fuel for gasoline blending [4]. According to the Renewable Fuels Association, the United States and Brazil produced 15,620 and 8260 million gallons of Biofuel ethanol in 2023, accounting for 53%and 28% of the world, respectively. Initially, first-generation fuel ethanol was primarily sourced from food crops, exacerbating concerns about competition for grains and land. Consequently, the advent of second-generation fuel ethanol aimed to mitigate the issues linked with the first generation, notably its reliance on food crops and fossil fuels [5]. The second generation fuel ethanol primarily utilizes lignocellulosic biomass as its raw material—a rich and renewable resource [6]. According to the Renewable Fuels Association, 50% of ethanol refineries operate using cellulosic feedstocks.

Abundant lignocellulosic species, including agricultural and forestry wastes like straw, dead grass, and forest branches, are potential raw materials for second-generation fuel ethanol production. Poplar, a fast-growing tree native to the northern hemisphere, stands as a principal constituent of man-made forests across northern and central China, with its distribution surpassing 66,667 km^2^ [7], yielding a significant volume of branches and processing waste annually [8]. However, the structural stability of poplar and the stress resistance of lignin to enzymes hinder its utilization [9]. Hence, a collaborative approach involving pretreatment, enzymatic digestion, and fermentation steps is needed to convert poplar to ethanol.

The objective of pretreatment is to disrupt the inherent structure of cellulose, enhance the accessibility of lignocellulosic substrates, and improve enzymatic digestion efficiency [10]. To achieve this objective, scientists have devised various pretreatment methods, primarily categorized as physical, chemical, physicochemical, and biological methods, all of which hold significant importance for lignocellulose utilization [11]. Among these methods, dilute sulfuric acid pretreatment stands out for its ability to break hydrogen bonds between poplar components, decrease lignocellulose crystallinity, and facilitate enhanced accessibility of cellulase to the substrate [12]. For instance, when treated with 0.1 g/L H_2_SO_4_, the enzymatic hydrolysis efficiency of bamboo shoots remarkably surged from 56% to 97% [13]. Furthermore, this method boasts simple operation, minimal risk, cost-effectiveness, and reduced corrosiveness to equipment, making it ideal for industrial scale-up and widely acknowledged as one of the most suitable approaches for large-scale applications [14]. Despite its merits, dilute acid pretreatment presents notable challenges, such as limited enzymatic hydrolysis efficiency stemming from incomplete lignin removal and the generation of inhibitors due to intense pretreatment conditions. Addressing these challenges, researchers have devised various pretreatment strategies to enhance enzymatic hydrolysis effectiveness. Among these, conventional alkaline pretreatment partially dissolves lignin, yet its efficiency remains modest. On the other hand, ionic liquid pretreatment shows promise in lignin removal but often entails significant hemicellulose loss [15,16]. Thus, to fully harness the potential of lignocellulosic components, employing a pretreatment strategy with robust and targeted lignin removal capabilities is pivotal for efficient cellulosic ethanol production. Sodium chlorite emerges as a highly efficacious alkaline oxidizing agent, demonstrating noteworthy efficacy in the elimination of lignin [17]. While the precise mechanism of its action remains elusive, the literature suggests that pretreatment utilizing diverse oxidation techniques can effectively deconstruct lignocellulosic structures [18]. Hence, it can decrease the lignin content within the substrate via oxidation, consequently mitigating the nonspecific adsorption of lignin onto cellulase. This reduction in lignin binding not only leads to a notable decrease in enzyme consumption and a shorter enzymatic hydrolysis duration but also augments the sugar yield. Notably, sodium chlorite exhibits a specific affinity towards lignin while exhibiting a high retention rate for both cellulose and hemicellulose [19]. This attribute substantially amplifies the sugar content within the hydrolysate.

The structural integrity of the raw material, treated with sodium chlorite, was partially disrupted. Furthermore, utilizing a gentler dilute acid pretreatment facilitates the partial removal of hemicellulose, thereby enhancing enzymatic hydrolysis efficiency. Integrating this process with enzymatic saccharification enables the degradation of both cellulose and residual hemicellulose in the raw material into fermentable monosaccharides, yielding hydrolysate with the high sugar concentration necessary for industrial production [20]. Subsequently, C6/C5 sugars can be co-utilized by microorganisms in fermentation reactions, wherein monosaccharides such as glucose and xylose are converted into ethanol through microbial metabolism [21]. This study delves into the potential of producing fuel ethanol from poplar biomass, thus broadening the horizon of second-generation ethanol production. The research is centered on developing an efficient process for converting poplar biomass into ethanol. This includes implementing a ‘lignin priority’ strategy, a ‘semi-xylose endpoint’ strategy, and employing a comprehensive optimization approach. Using the early-stage construction of an *S. cerevisiae* strain capable of efficient C6/C5 co-fermentation, the study establishes an effective biomass ethanol process for poplar wood, addressing preliminary challenges within the production technology pathway for second-generation fuel ethanol.

## 2. Materials and Methods

### 2.1. Materials

The poplar powder underwent screening through a 40-mesh sieve. Cellic CTec3 cellulase (Novozymes A/S, Bagsværd, Denmark), with an activity of 115.0 filter paper units (FPU)/mL, was utilized in this study. Liquid chromatography standard drugs were procured from Sigma-Aldrich LLC (St. Louis, MI, USA). Sulfuric acid (98%), calcium oxide, and sodium chlorite (80%) were obtained from Sinopatharm Chemical Reagent Co., Ltd. (Shanghai, China).

### 2.2. Delignification Process

SC solution with different mass fractions was prepared for delignification pretreatment. Poplar samples (3.0 g) were treated with 30 mL of 2%–8% (*w*/*v*) SC at 70 °C for 0.5–6 h [22]. The delignification process took place in a glass reactor, with temperature control provided by a water bath. After delignification, the residual SC solution was collected through vacuum-assisted filtration and then washed to achieve neutrality. Following this, analysis of the poplar components was conducted using methods established by the National Renewable Energy Laboratory (NREL) [23]. The crude protein content was determined using the KN method [24]. Starch content was assessed with the BOXBIO starch content kit. Glucose, xylose, and arabinose were analyzed using NREL methodology.

### 2.3. Dilute Sulfuric acid Pretreatment and Enzymatic Hydrolysis

The delignified poplar was pretreated with dilute sulfuric acid in a conical flask, and the delignified poplar (3 g) and DSA (0–1%) were pretreated at a solid–liquid ratio of 1:10 for 2 h at 121 °C [25].

The dilute sulfuric acid pretreatment product does not require washing. The pH was adjusted to 4.8 with calcium oxide, and the product was enzymatically digested in a conical flask (100 mL) at 50 °C. In order to investigate the effect of cellulase dosage and reaction time on enzymatic hydrolysis, the dosage of cellulase was 10–35 FPU/g (dry matter) and the reaction time was 24–48 h. When the solid load of poplar was 20% (*w*/*v*), 10 U/g (dry matter) *β*-glucosidase was added for enzymatic hydrolysis.

The sugar yield was calculated by the following formula:Yield_Sugar_ (%) = (C_HPLC_ × V)/(m × C_Sugar_) × 100 
where C_HPLC_ is the total concentration of glucose, xylose, and arabinose in the hydrolyzate, g/L; V is the volume of enzymatic digestion, L; m is the total amount of poplar material, g; C_Suger_ is the percentage of glucose, xylose, and arabinose in poplar, %.

### 2.4. Fermentation and Strains

The enzymatic hydrolysate was used for fermentation at 30 °C for 48 h [26]. The fermentation was performed in an oxygen-limited flask (100 mL) with a shaker at 200 rpm. The inoculated yeast was 3.5 OD_600_ (0.5 g DCW/L). When using the fermenter, the filling volume was 70–75% of the total volume of the fermenter. The aeration volume was 0.6 vvm during the first 6 h of fermentation. An appropriate amount of antifoam was added during the initial fermentation.

Strains LF1 and 6M-15 are two engineered strains developed by our laboratory. Strain LF1 is an industrial strain with efficient co-utilization of glucose and xylose, which was constructed in our previous work. 6M-15 is a mutant strain derived from LF1 and is suitable for highly toxic hydrolysate [27]. Strains 6M-151 and 6M-156 were derived from 6M-15 mutation and adaptive evolution.

### 2.5. Analytical Methods

The filter paper enzyme activity (FPU) was determined by the dinitrosalicylic acid method (DNS) method [28]. *β*-Glucosidase and *β*-Xylosidase activities were determined by measuring *β*-nitrophenol release [29]. Monosaccharides and ethanol were measured by high-performance liquid chromatography (Waters e2695, Milford, MA, USA) and refractive index detector (Waters 2414, USA), using an Aminex Bio-Rad HPX-87H column (300 mm × 7.8 mm) and a refractive index detector, together with 5 mM H_2_SO_4_ as the mobile phase at a flow rate of 0.5 mL/min at 45 °C [27]. Other substances were measured by GL Sciences WondaSil C18 column (4.60 mm × 250 mm, 5 μm) and UV Detector (Waters, 2998 PDA Detector, USA) [30].

The formulas were used for the estimation of ethanol yield, the ratio of ethanol yield to the theoretical value, and the energy transformation ratio:Yield_Ethanol_ (g/g) = C_Ethanol_/C_Suger_

Ratio_Ethanol_ (%) = Yield_Ethanol_/0.51 × 100
Energy transformation ratio (%) = G_Ethanol_ × E_Ethanol_/G_Bio_/E_Bio_
where C_Ethanol_ is the ethanol concentration at the end of fermentation, g/L; C_Suger_ is the glucose and xylose concentration at the beginning of fermentation, g/L; and 0.51 is the value of 1 g of glucose or xylose that could theoretically be converted to ethanol; G_Ethanol_ is the ethanol weight, Kg; E_Ethanol_ is the calorific value of ethanol, MJ/Kg; G_Bio_ is the biomass weight, Kg; E_Bio_ is the calorific value of biomass.

## 3. Results and Discussion

### 3.1. Chemical Compositions of Poplar

The primary constituents of poplar biomass encompass cellulose, hemicellulose, and lignin, collectively constituting over 85% of their composition (Table 1). The residual components include extracts, along with modest quantities of crude protein and ash. Noteworthy is the total carbohydrate content, which stands at 60.79%, predominantly sourced from cellulose (glucose) and hemicellulose (xylose). Compared with other lignocellulosic feedstocks, poplar biomass has a higher content of cellulose and lignin. For example, the cellulose of corn cob, rice straw, and barley straw is 27.71%, 32.15%, and 37.60%, respectively, and the lignin of corn straw, corn cob, and barley straw is 11.70%, 9.4%, and 15.80%, respectively [31]. Moreover, the hemicellulose within poplar biomass primarily comprises xylan, with minimal arabinose presence [32]. Presently, recombinant *S. cerevisiae* strains showcase remarkable co-fermentation capabilities for both glucose and xylose, enabling efficient utilization of the entire poplar biomass. Approximately 70% of carbohydrates are accessible for fermentation, consistent with prior research findings [33].

### 3.2. Delignification Pretreatment of Poplar with Sodium Chlorite

Lignin serves as a primary component of plant cell walls, and its structural units impede the degradation of cellulose [31]. Concurrently, certain compounds, such as phenolics, formed during lignin degradation, exert inhibitory effects on microorganisms, thereby influencing the rate and yield of ethanol fermentation. However, these effects can be mitigated through appropriate pretreatment [34]. Sodium chlorite finds extensive application as a bleaching agent in the paper and textile industries [35]. Several studies have suggested that sodium chlorite can also act as a potent oxidant, capable of breaking down lignin [20].

The changes in the three primary components of the raw material treated with sodium chlorite are shown in Table 2. It is evident that an increase in sodium chlorite dosage from 2% to 8% leads to a significant rise in lignin removal rate; however, it also results in gradual loss of cellulose and hemicellulose. At an 8% (*w*/*v*) sodium chlorite loading, the removal rate of lignin reaches 80.52%. Nevertheless, due to stable chemical bonding between lignin–carbohydrate complexes, even with the addition of 8% (*w*/*v*) sodium chlorite, residual lignin remains consistent with previous experimental findings [36]. Notably, at a 5% (*w*/*v*) sodium chlorite concentration, delignification efficiency exhibits a gradual increase. Furthermore, at a 6% (*w*/*v*) sodium chlorite concentration, there is a significant reduction in hemicellulose content, contrary to our expectation of retaining cellulose and hemicellulose to the greatest extent possible. Balancing production costs and waste treatment requirements and ensuring effective delignification outcomes suggest that the optimal sodium chlorite dosage stands at 5% (*w*/*v*).

To further explore the impact of SC pretreatment duration on its efficacy, we conducted tests with varying retention times (Table 3). Notably, pretreating with sodium chlorite at 70 °C for 0.5 h resulted in a mere 28.74% removal rate of lignin. As the retention time increased, the lignin removal rate from SC pretreatment escalated from 28.74% to 74.69%. Concomitantly, there was an increase in cellulose and hemicellulose loss from 0.84% and 2.37% to 3.37% and 8.09%, respectively. After a pretreatment duration of 3 h, there was scarcely any further increase in lignin removal rate, with only a marginal 1.60% increment observed between 3 and 6 h of treatment time. Additionally, at the 3 h mark of SC pretreatment, cellulose and hemicellulose loss remained below five percent, signifying minimal carbohydrate loss during this processing step. Generally, sodium chlorite’s optimal temperature range for pretreatment falls between 60 and 80 °C; however, changes in temperature exert limited effects on reaction rates due to sodium chlorite’s propensity to decompose at higher temperatures [22,37,38,39]. In summary, the pretreatment method involving 10% (*w*/*v*) solid loading, 5% (*w*/*v*) SC, and a retention time of 3 h at 70 °C was selected.

It is noteworthy that compared to alkaline pretreatment, which exhibited a low lignin removal rate, SC pretreatment led to a significant reduction in lignin content by 73.09% [40]. Furthermore, sodium chlorite pretreatment showcased superior retention of cellulose and hemicellulose compared to eutectic solvent pretreatment while achieving a high lignin removal rate [41]. Following delignification, the proportion of cellulose and hemicellulose in poplar increased to 49.78% and 21.26%, respectively.

### 3.3. Depolymerization of Delignified Poplar by Sulfuric Acid and Cellulase

Following this, direct digestion of delignified poplar by cellulase yielded low efficiency. Semi-fibers can be efficiently degraded through sulfuric acid pretreatment, facilitating enzymatic sugar extraction [42]. However, excessive use of this technique may lead to adverse effects such as sugar loss, inhibitor production, and equipment corrosion [43]. Additionally, partial depolymerization of xylan may occur due to the presence of xylanase in cellulase [44]. Hence, we contend that striving for the highest xylose yield during pretreatment is unnecessary. Instead, we introduced the concept of a ‘hemixylose endpoint,’ denoting the pretreatment intensity at which the xylose yield reaches about half its theoretical value. This approach allows for cost reduction in acid pretreatment and minimizes inhibitor production.

In this study, DSA pretreatment was conducted at 120 °C with a retention time of 120 min, using a range of H_2_SO_4_ loadings 0–1.0% (*w*/*v*). The changes in glucose and xylose yields are depicted in Figure 1A. As the acid loading increased from 0 to 1.0%, the xylose yield rose from 10.89% to 76.66%. However, at a 0.4% H_2_SO_4_ loading, the hydrolysis efficiency of the acid treatment gradually declined. Considering this, two sets of acid loadings higher and lower than 0.4% were chosen for enzymatic hydrolysis tests. Enzymatic hydrolysis was conducted for 48 h using cellulase supplemented with 35 FPU/g (dry weight of substrate). At a 0.4% H_2_SO_4_ loading, the maximum glucose and xylose yields reached 51.21 g/L and 23.07 g/L, respectively, with sugar yields of 92.60% and 95.49%, and a total sugar yield of 93.48%. However, increasing the H_2_SO_4_ loading to 0.6% did not result in higher glucose and xylose production; instead, there was a slight decrease. We hypothesized that a stronger dilute acid pretreatment intensity might promote the conversion of more glucose and xylose into inhibitors such as 5-HMF and furfural [43].

Through the described process, the yields of glucose and xylose tended to stabilize above 90%, as illustrated in Figure 1B. Interestingly, even as the cellulase load was gradually reduced (10–35 FPU/g), the sugar concentration and yield showed no significant decrease. Even at a cellulase load of 10 FPU/g, the sugar yield decreased by only 6.08% (Figure 2A). This suggests that the combination of SC and DSA pretreatment significantly enhanced the enzymatic hydrolysis efficiency. At a cellulase load of 35 FPU/g, glucose and xylose concentrations reached 51.21 g/L and 23.07 g/L, respectively, with yields of 95.09% and 97.83%, respectively. Concurrently, the total sugar yield peaked at 93.48%. However, the concentrations of inhibitors (acetic acid, furfural, 5-HMF, and phenols) in the hydrolysate reached 3.61 g/L, 0.39 g/L, 1.37 g/L, and 1.63 g/L, respectively. Notably, sugar concentration and yield increased gradually with cellulase loading above 15 FPU/g.

Moreover, the impact of reaction time on enzymatic hydrolysis was investigated. Clearly, enzymatic hydrolysis time significantly influenced sugar yield. For instance, with an enzyme load of 35 FPU/g, and a 12 h reaction, glucose and xylose yields were 61.74% and 83.81%, respectively. Doubling the enzymatic hydrolysis time increased the yields to 85.65% and 90.45%, respectively. By extending the time to 36 h, these proportions escalated to 92.74% and 95.21%, respectively (Figure 2B). However, beyond 36 h, sugar concentration and yield plateaued, indicating that the enzymatic hydrolysis process becomes saturated.

The cost of cellulase constitutes the major portion of second-generation fuel ethanol production expenses, and minimizing cellulase usage can effectively lower process costs [44]. To achieve cost reduction while attaining satisfactory monosaccharide yields, Response Surface Methodology (RSM) was employed to optimize dilute acid pretreatment and enzymolysis parameters. Specifically, three independent variables, namely H_2_SO_4_ loading (A), cellulase loading (B), and enzymolysis time (C), were determined. The upper and lower limits of these experimental parameters are detailed in Table 4, with total sugar yield (glucose and xylose) selected as the response value. The Box–Behnken Design (BBD) facilitated 17 experimental runs with varying values for the three variables.

At the H_2_SO_4_ loading of 0.4%, a cellulase dosage of 10 FPU/g, and an enzymatic hydrolysis time of 48 h, the maximum total sugar yield reached 86.28%, corresponding to glucose and xylose concentrations of 49.55 g/L and 20.59 g/L, respectively (Table 4). Leveraging the data from these 17 experimental groups, a second-order polynomial equation was derived to predict the total sugar yield. The regression equation for total sugar yield is calculated as follows: Total sugar yield = +77.11 + 15.32×H_2_SO_4_ loading + 8.67 × Cellulase loading + 5.07 × Time − 2.62 × H_2_SO_4_ loading×Cellulase loading − 0.11 × H_2_SO_4_ loading × Time − 2.26×Cellulase loading × Time − 9.97 × H_2_SO_4_ loading^2^ − 3.54 × Cellulase loading^2^ − 1.63 × Time^2^. The positive and negative symbols associated with each variable denote synergy and antagonism, respectively.

The statistical analysis technique utilized in this study was Quadratic ANOVA. The significance of the model is highlighted by an F-value of 318.90 and a *p*-value of less than 0.0001, indicating a confidence level of 99.99%. Table 5 further identifies significant factors influencing monomer sugar yield as A, B, C, AB, BC, A^2^, B^2^, and C^2^. Although the impact of the AC combination is relatively minor, it plays a crucial role in supporting the hierarchical model. By employing software prediction, optimal conditions were determined, resulting in a monomer sugar yield of 87.31% of the theoretical maximum. The actual monosaccharide yield obtained under these optimal pretreatment conditions closely approximated the predicted value, reaching 87.67%.

Total sugar yield was utilized to analyze the effects of factors on hydrolysis reaction, with H_2_SO_4_ quantity, enzyme dosage, and hydrolysis time being the primary factors under investigation (Figure 3). The interaction between H_2_SO_4_ and enzyme loading exhibited the least influence on monomer sugar yield. Notably, the density profile on the H_2_SO_4_ loading axis surpassed that on the enzyme dosage axis, suggesting a more substantial impact of H_2_SO_4_ on the reaction. Increased H_2_SO_4_ loading effectively removes hemicellulose, augmenting the specific surface area for cellulose and enzyme interaction with the substrate [45]. Previous studies have reported a strong correlation between xylan removal and digestibility [46].

The optimization of sugar yield in this study achieved 87.48%, slightly below the highest sugar yield of 93.48% obtained under identical experimental conditions. However, the mild pretreatment process significantly reduced the production of inhibitors. Specifically, the contents of acetic acid, furfural, 5-HMF, and phenols were 5.98%, 25.64%, 18.25%, and 5.52%, respectively. Furthermore, enzyme usage was reduced by approximately 67%, markedly improving the process’s economic viability.

### 3.4. Evaluation of Yeast Fermentation of Poplar Biomass Hydrolysates

The optimized hydrolysate presents a complex composition, containing not only glucose and xylose but also certain inhibitors, this is a drawback associated with dilute acid pretreatment [47]. According to the literature reports, if the concentration of acetic acid exceeds 5.0 g/L or that of furfural surpasses 0.2 g/L, it may exert adverse effects on the growth and fermentation of *S cerevisiae* [48]. Furthermore, in the presence of multiple inhibitors, their interaction can significantly amplify their toxicity, resulting in heightened inhibition of microbial growth and fermentation compared to any single inhibitor [34].

Ethanol fermentation was conducted using poplar as a substrate, employing a second-generation fuel ethanol production industrial strain LF1, specifically developed and optimized in the laboratory. LF1, obtained through genetic modification and adaptive evolution, possesses notable characteristics such as xylose utilization capability and resistance to inhibitors. Its ability to efficiently utilize both xylose and glucose is particularly noteworthy, positioning LF1 as a promising candidate for practical second-generation bioethanol production [30].

However, fermentation with strain LF1 exhibited significant inhibition in growth and metabolism, failing to meet expectations regarding xylose utilization rate, cell density, and ethanol yield. Analysis of the fermentation broth composition at various stages revealed a gradual increase in acetic acid concentration and a corresponding decrease in pH with fermentation progression, consistent with prior literature [49]. The microbial production of acetic acid during fermentation can detrimentally impact yeast growth and fermentation activity, particularly in later fermentation stages when acetic acid and other toxic compound concentrations are elevated [50].

With an acetic acid dissociation constant (pKa) of 4.75, indicating a predominance of undissociated acetic acid molecules at pH values below pKa, the lipophilic nature of undissociated acetic acid facilitates its diffusion across the cytoplasmic membrane. Upon dissociation inside the cell, it acidifies the intracellular environment, hindering growth and metabolism [51]. Hence, our aim was to alleviate the inhibitory effects of acetic acid by adjusting the fermentation broth pH, thereby enhancing fermentation outcomes. We investigated the impact of hydrolysates with varying pH levels on *S. cerevisiae* fermentation by adjusting the hydrolysate pH to 4.5, 5.0, and 5.5 using a CaO reagent.

Figure 4A illustrates that the glucose utilization of strain LF1 remained largely unaffected by pH variations. However, pH exerted a significant influence on xylose utilization. At pH 4.5, the xylose utilization rate was merely 66.97%. Upon adjusting the pH to 5.0, the xylose utilization rate surged to 94.45%. Remarkably, at pH 5.5, xylose utilization peaked at 100% (Figure 4B). Concurrently, Figure 4C demonstrates that adjusting the pH from 4.5 to 5.0 resulted in the ethanol yield, calculated based on the total sugar concentration of the hydrolysate, reaching 90.64% of the theoretical value. Nevertheless, when the pH was elevated to 5.5, ethanol production remained largely unchanged, possibly due to a higher carbon source allocation towards cell growth, thereby impacting ethanol production (Figure 4D).

To comprehensively assess the industrial production potential of this process route, a 50 L batch fermentation was conducted using the strain LF1, incorporating a 10% solid poplar hydrolysate concentration. The maximum ethanol yield achieved through the 50 L batch fermentation amounted to approximately 0.47 g/g of fermentable sugar, equivalent to 93.13% of the theoretical yield (Figure 5). Compared to shake-bottle fermentation, this process not only reaches production goals more expeditiously but also attains a higher ethanol yield. This improved performance may be attributed to enhanced control over fermentation conditions, enabling a greater proportion of fermentable sugars to be directed toward ethanol fermentation rather than cell growth. These outcomes affirm the stability of the ethanol pathway from poplar biomass as the fermentation scale increases. Remarkably, achieving a 50 L fermentation scale represents the highest scale to date for cellulosic ethanol production from poplar biomass, offering valuable technical insights for industrial-scale ethanol production from poplar biomass.

Moreover, we conducted an energy conversion rate analysis for poplar biomass ethanol to evaluate its competitiveness and market potential. This analysis is instrumental in assessing the economic impact of achieving specific conversion performance targets, thereby guiding efforts to reduce process costs. Under ideal conditions, where processed waste is incinerated and waste heat is recovered to offset processed energy requirements, the energy conversion rate of poplar biomass ethanol reaches 37.67%. This figure surpasses the energy conversion rate of cellulosic ethanol [52,53,54,55]. Finally, after delignification pretreatment, dilute sulfuric acid pretreatment, and enzymolysis, 100 g raw poplar obtained 41.52 g glucose, 15.53 g xylose, which was fermented to produce 26.81 g ethanol (Figure 6).

### 3.5. Effect of 20% Solid Loading Delignified Poplar Hydrolysis Products Ethanol Fermentation

Sufficient data confirm that pretreated poplar biomass exhibits favorable enzyme digestibility and ethanol yield at 10% solid loading. However, for an economically viable distillation process, ethanol concentration in the fermentation solution typically needs to reach up to 5% (*v*/*v*) [56]. Therefore, increasing the solid load in enzymatic hydrolysis is necessary to attain a higher sugar concentration for ethanol fermentation [57]. Nonetheless, several technical challenges impede the widespread application of enzymatic hydrolysis with high solid loads. For instance, in the enzymatic hydrolysis of lignocellulose, higher sugar concentrations often lead to a significant reduction in the initial enzymatic hydrolysis rate and final sugar yield. Moreover, the accumulation of inhibitors generated during the pretreatment process exacerbates the hydrolysate’s toxicity [58].

Given that our dilute acid pretreatment and enzymolysis form an integrated process, we opted for a pretreatment condition with a low sulfuric acid load (0.4% H_2_SO_4_ load, 120 min at 120 °C) while increasing the solid load of delignified poplar to 20%. Following pretreatment and subsequent pH adjustment to 4.8 with the addition of 35 FPU/g cellulase, enzymatic hydrolysis efficiency reached only 69.66% at 48 h (Table 6). Concurrently, we observed higher cellobiose levels in the hydrolysate, indicative of incomplete cellulose hydrolysis [59]. Due to the elevated solid load, current pretreatment and enzymatic hydrolysis conditions are suboptimal. However, intensifying sulfuric acid pretreatment to enhance hydrolysis exacerbates inhibitor production, contrary to our objectives. Previous studies have shown that *β*-glucosidase effectively alleviates cellobiose feedback inhibition, thereby enhancing enzymatic hydrolysis efficiency [60]. Through the addition of 10 U/g *β*-glucosidase, enzymatic hydrolysis efficiency increased to 91.36% (Table 6).

At 20% solid load (*w*/*v*), the primary hydrolyzed products comprise glucose (101.66 g/L), xylose (45.50 g/L), acetic acid (6.92 g/L), furfural (0.73 g/L), 5-HMF (2.53 g/L), and phenols (3.06 g/L). In this study, strains LF1 and 6M-15, capable of co-fermenting glucose and xylose, were utilized to evaluate the fermentation performance of hydrolysates with a 20% solid load. Strain 6M-15, derived from strain LF1 through mutagenesis and adaptive training, exhibits a more balanced xylose utilization ability and robustness than LF1 [27].

As depicted in Figure 7A,B, glucose is almost completely consumed by 36 h, while strains LF1 and 6M-15 have only utilized 37.00% and 42.40% of xylose, respectively, by this time. Notably, at 48 h, 6M-15 achieves higher ethanol yields than LF1, with a yield of 0.39 g/g. The sugar (glucose and xylose) consumption patterns by the two strains align with their inherent metabolic characteristics, as demonstrated in the experiment with 20% solid loading hydrolysate. It is anticipated that the metabolic capacity of the strains plays a crucial role in determining hydrolyzed sugar consumption, especially considering the higher inhibition levels at a 20% solid loading. The inhibition of xylose consumption in the hydrolysate may be attributed to the presence of multiple inhibitors, which exert minimal impact on glucose consumption for both strains [27]. In summary, the fermentation performance of 6M-15 on 20% solid loading hydrolysate surpasses that of LF1, albeit both strains are constrained by the effects of the hydrolysate.

Activated carbon detoxification is reported as an effective method for removing fermentation inhibitors while minimally reducing fermentable sugars, making it a more practical option for industrial applications [61]. Resin detoxification also exhibits a favorable inhibitor removal effect, particularly targeting weak acids, furan aldehydes, and phenols, and boasts renewable characteristics [62]. However, drawbacks exist in the detoxification process. Although resin detoxification is effective, it often leads to significant loss of fermentable sugars [63]. Conversely, activated carbon detoxification results in minor sugar loss but offers a less potent inhibitor removal effect compared to resin. It is important to note that activated carbon is a disposable product, and its use in the detoxification process leads to a reduction in hydrolysate volume. Additionally, increasing detoxification steps may introduce process complexity and costs, which are undesirable. Therefore, enhancing fermentation efficiency of hydrolyzed products through strain customization is preferable. We employed the ARTP mutagenesis technique and adaptive training to select strains based on the strain 6M-15 starting strain, utilizing a 20% solid loading hydrolysate pressure medium.

The results, as demonstrated in Figure 8A,B, indicate that strain 6M-156 exhibited the highest xylose utilization rate (58.49%), consuming 38.01% more xylose than the original strain. 6M-151 yielded the highest ethanol concentration, reaching 80.19% of its theoretical value, representing a 5.10% increase over the original strain’s yield. The fermentation performance of 6M-151 on 20% solid loading hydrolysate surpassed that of 6M-15; however, its full potential remains unrealized. In conclusion, the fermentation performance of 6M-156 on hydrolysate with a 20% solid loading was commendable, demonstrating a similar ethanol yield to 6M-151 and superior xylose utilization ability. We believe that customizing strains for specific hydrolysates can further enhance fermentation performance. Reasonable strain modifications hold significant potential for substantial improvements.

## 4. Conclusions

Combined with pretreatment and enzymolysis, the ‘lignin priority’ and ‘semi-xylose endpoint’ strategy obtained a higher saccharification rate of 93.48%. Using Box–Behnken design optimization, the most economical and feasible parameters were obtained, and the saccharification rate was 87.67%. Through the implementation of accurate process control, using poplar hydrolysis products as a medium, and combining with the second generation fuel ethanol strain LF1 fermentation of 50 L scale, ethanol production reached 35 g/L. The ethanol yield accounted for 93.13% of the theoretical value, reaching the highest level reported so far. It is proven that the process has the advantages of a high saccharification rate and high ethanol yield at the same time. In addition, the hydrolysis conditions of 20% solid loading delignified poplar were optimized, and the saccharification rate reached 91.36%, which solved the problem of low enzymatic hydrolysis efficiency under high solid load. And the strain suitable for poplar hydrolysate was selected by irrational method, the xylose utilization rate increased by 38.01%, and ethanol yield increased by 5.10%. This study provides a potential process route for the production of cellulosic ethanol from poplar biomass and provides a new idea and opportunity for improving the energy utilization value of poplar biomass in ethanol production.

## Figures and Tables

**Figure 1 microorganisms-12-01174-f001:**
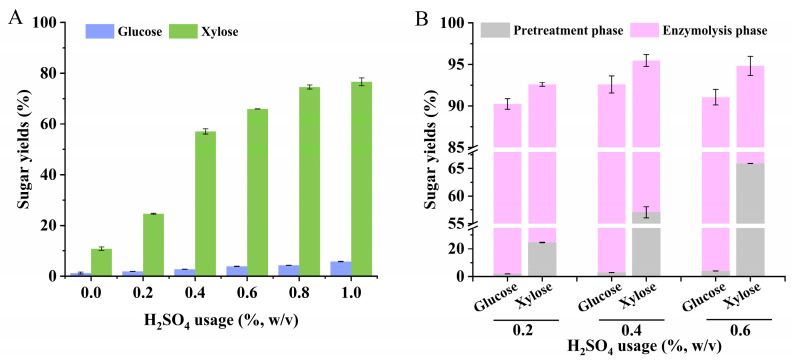
Sugar production rates of glucose and xylose after pretreatment (**A**) and enzymatic hydrolysis (**B**) with different acid loads of H_2_SO_4_.

**Figure 2 microorganisms-12-01174-f002:**
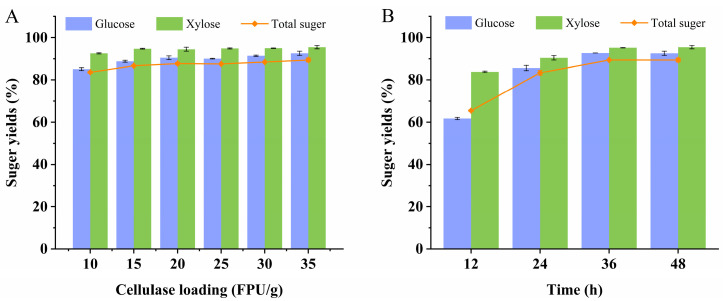
The comparison of yields of glucose, xylose, and total sugar via enzymatic hydrolysis of cellulase load and time. (**A**) The effect of cellulase load on the sugar yield, and (**B**) the effect of time on the sugar yield.

**Figure 3 microorganisms-12-01174-f003:**
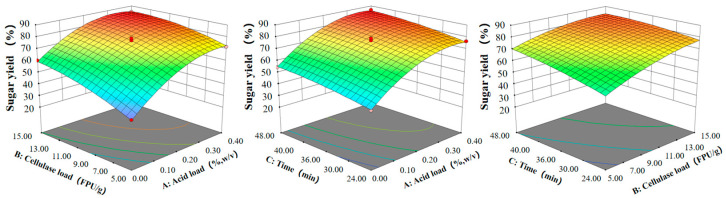
Response surfaces presenting the optimal parameters for effects of combined pretreatment. The more red the color, the higher the sugar yield, the more blue the color, the lower the sugar yield.

**Figure 4 microorganisms-12-01174-f004:**
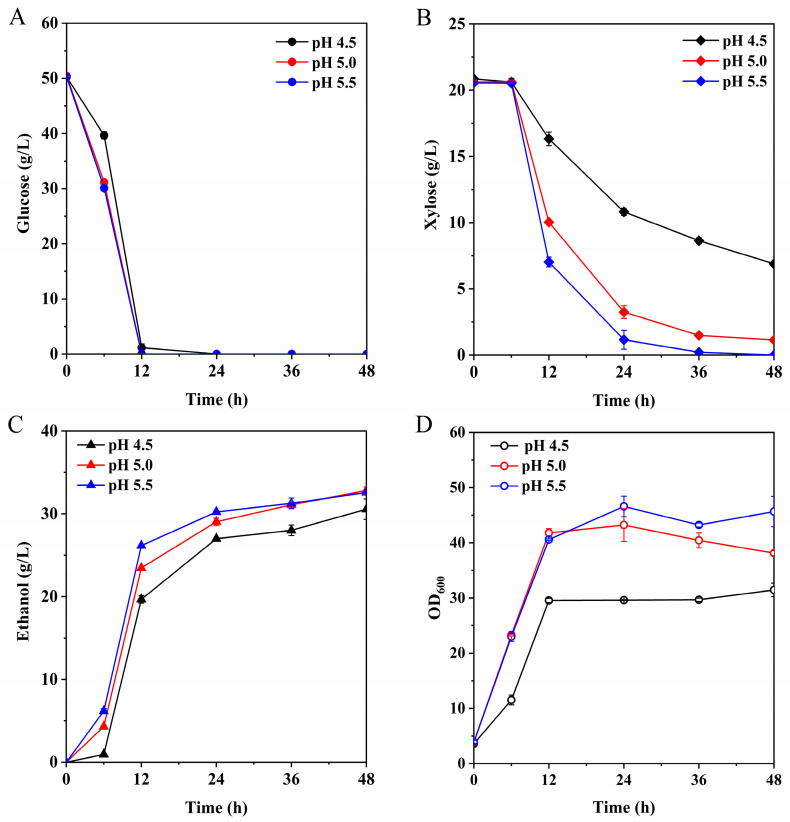
Trends of glucose (**A**), xylose (**B**), ethanol (**C**) and cell density (**D**) neutralized in fermentation broth at different pH.

**Figure 5 microorganisms-12-01174-f005:**
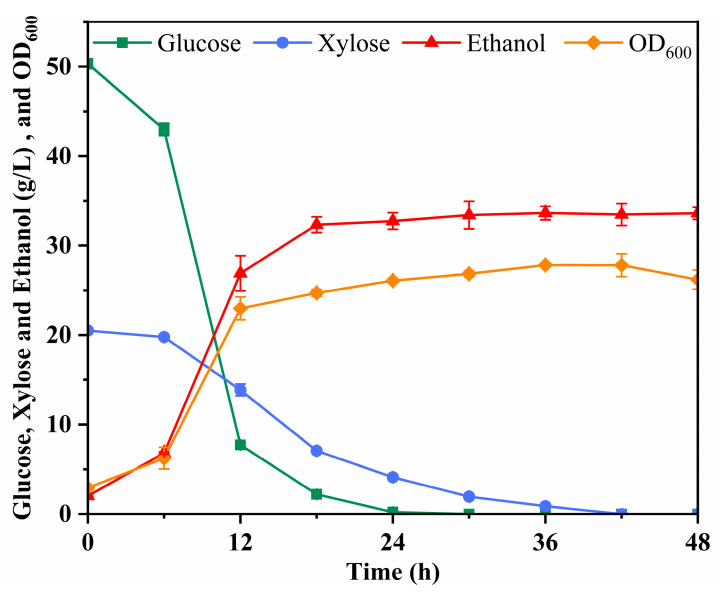
Evaluation of fermentation results of 50 L scale-up experiments.

**Figure 6 microorganisms-12-01174-f006:**
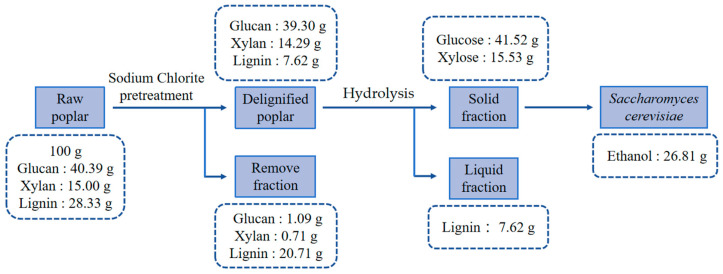
Mass balance in preparing hydrolysate and producing ethanol from poplar.

**Figure 7 microorganisms-12-01174-f007:**
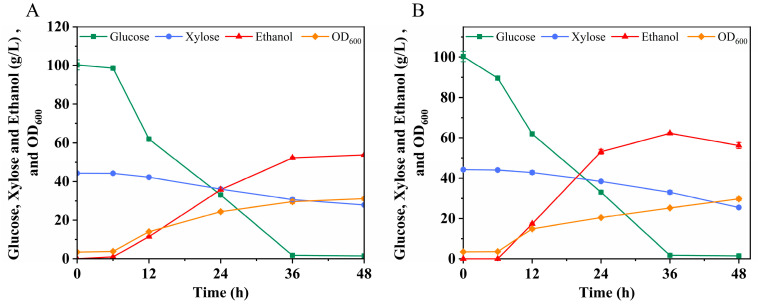
Evaluation of fermentation of strain LF1 (**A**) and 6M-15 (**B**) in 20% solid load hydrolysate.

**Figure 8 microorganisms-12-01174-f008:**
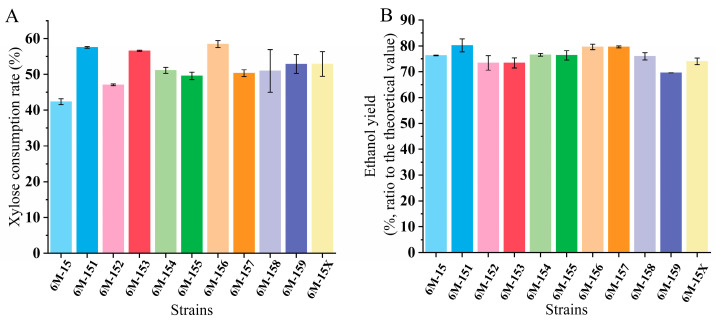
Xylose utilization rate (**A**) and ethanol yield (**B**) of strain screening.

**Table 1 microorganisms-12-01174-t001:** Composition of poplar (dry basis).

	Components	Contents (%)
Biomass component	Cellulose	40.39 ± 1.54
	Hemicellulose	15.00 ± 0.54
	Protein	0.42 ± 0.01
	Lignin (acid-soluble)	1.64 ± 0.05
	Lignin (acid-insoluble)	26.69 ± 0.23
	Ash	1.05 ± 0.03
	Extractives	8.21 ± 0.01
	Glucose	44.88 ± 1.71
Major monosaccharide	Xylose	17.05 ± 0.61
	Arabinose	0.09 ± 0.01

**Table 2 microorganisms-12-01174-t002:** Solid yields and component removal ratio of samples after sodium chlorite mass fraction.

Levels (%)	Solid Yield (%) *	Removal Ratio (%)		
		Cellulose	Hemicellulose	Lignin
2	90.86 ± 0.49	0.91 ± 0.09	2.24 ± 0.14	31.33 ± 0.28
3	86.37 ± 1.29	1.38 ± 0.67	3.34 ± 0.03	45.08 ± 0.21
4	82.66 ± 0.78	2.03 ± 0.25	4.45 ± 0.32	57.61 ± 0.72
5	80.31 ± 0.49	2.69 ± 0.67	4.72 ± 0.29	73.09 ± 0.19
6	79.74 ± 1.83	2.82 ± 0.39	6.09 ± 0.42	76.44 ± 1.62
7	78.39 ± 1.02	3.01 ± 0.17	7.54 ± 0.29	78.69 ± 0.78
8	77.80 ± 1.16	3.47 ± 0.21	8.30 ± 0.54	80.52 ± 1.23

* The ratio of the solid mass of poplar after treatment to that before treatment.

**Table 3 microorganisms-12-01174-t003:** Solid yields and component removal ratio of samples after sodium chlorite pretreatment time.

Levels (h)	Solid Yield (%) *	Removal Ratio (%)		
		Cellulose	Hemicellulose	Lignin
0.5	93.88 ± 1.29	0.84 ± 0.14	2.37 ± 0.17	28.74 ± 1.19
1	90.23 ± 0.65	0.99 ± 0.03	2.84 ± 0.21	43.28 ± 1.07
2	83.47 ± 0.63	1.11 ± 0.46	2.96 ± 0.54	59.77 ± 0.36
3	80.31 ± 0.49	2.69 ± 0.67	4.72 ± 0.29	73.09 ± 0.19
4	79.66 ± 0.84	2.82 ± 0.61	5.77 ± 1.1	72.06 ± 2.10
5	79.82 ± 0.40	3.08 ± 0.25	6.86 ± 0.38	73.98 ± 0.15
6	78.41 ± 0.24	3.37 ± 0.00	8.09 ± 0.13	74.69 ± 0.74

* The ratio of the solid mass of poplar after treatment to that before treatment.

**Table 4 microorganisms-12-01174-t004:** The Box–Behnken design was used to optimize the response value of the hydrolysis experiment of delignification feedstock.

RUN	H_2_SO_4_ Loading (%, *w*/*v*)	Cellulase Loading (FPU/g)	Time (h)	Glucose (g/L)	Xylose (g/L)	Total Sugar Yield (%)
1	0.4	5	36	40.02	18.50	71.98
2	0.4	10	24	43.42	18.54	76.57
3	0.2	5	48	39.11	18.19	71.02
4	0.2	5	24	29.18	16.22	56.13
5	0.2	10	36	43.47	18.80	76.37
6	0.2	10	36	43.53	18.63	77.02
7	0.4	15	36	48.78	20.00	84.70
8	0.2	10	36	44.68	19.18	78.83
9	0.4	10	48	49.55	20.59	86.28
10	0.2	10	36	43.46	18.69	76.22
11	0	5	36	18.64	11.63	37.27
12	0	15	36	34.01	14.92	60.47
13	0	10	48	30.03	14.21	54.68
14	0	10	24	23.38	12.52	44.53
15	0.2	15	24	43.76	18.74	77.39
16	0.2	15	48	47.57	19.98	83.22
17	0.2	10	36	43.62	18.61	77.12

**Table 5 microorganisms-12-01174-t005:** Analysis of Variance (ANOVA) for the optimization of hydrolyzed sugar yield from delignified poplar.

Source	Sum of Squares	df	Mean Square	F-Value	Prob > F	Remark
Model	3245.80	9	360.64	318.90	<0.0001	S
A- H_2_SO_4_ loading	1878.23	1	1878.23	1660.80	<0.0001	S
B- Cellulase loading	601.70	1	601.70	532.04	<0.0001	S
C-Time	205.84	1	205.84	182.01	<0.0001	S
AB	27.46	1	27.46	24.28	0.0017	S
AC	0.048	1	0.048	0.043	0.8420	nS
BC	20.52	1	20.52	18.15	0.0037	S
A^2^	418.19	1	418.19	369.78	<0.0001	S
B^2^	52.79	1	52.79	46.68	0.0002	S
C^2^	11.20	1	11.20	9.90	0.0162	S
Residual	7.92	7	1.13			
Lack of Fit	3.61	3	1.20	1.12	0.4408	nS
Pure Error	4.31	4	1.08			
Cor Total	3253.72	16				
R^2^	0.9976					

S: significant; nS: not significant; df: degree of freedom.

**Table 6 microorganisms-12-01174-t006:** Sugar yields from cellulolytic hydrolysis after increased solid loading.

Source	Glucose Yield (%)	Xylose Yield (%)	Total Sugar Yield (%)
Enzymolysis	62.80 ± 1.23	88.24 ± 0.87	69.66 ± 0.97
Enzymolysis+*β*-Glucosidase	89.45 ± 2.74	94.16 ± 1.54	91.36 ± 2.27

## Data Availability

Data are contained within the article.

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
