# Peer review of "Highly Efficient Production of Cellulosic Ethanol from Poplar Using an Optimal C6/C5 Co-Fermentation Strain of Saccharomyces cerevisiae"

_microorganisms, 2024, doi:10.3390/microorganisms12061174_

Round 1
Reviewer 1 Report
Comments and Suggestions for Authors
Dear Authors,
I read your manuscript with a huge interest since it touches on crucial problems of 2nd generation bioethanol creation. I’m impressed by the results presented, like 35 g of ethanol /1L at 10% of solid loading. I consider Your manuscript as valuable to be published, but to fit the journal demands, some improvements should be made – please see my detailed comments below:
Line 15 in the Abstract and apropriate parts with this context in the whole manuscript: ethanol yield equal to 35 g/L was obtained at 10% of substrate loading (approximately 40% of cellulose and 15% of hemicellulose -> 55g/100g of polymerised sugars) thus I think that the authors should clarify somethere in the text if the solid load was „crude solid=raw material” or deligified solid load ? since it is crucial for the proper yield results evaluation.
Lines 26-28: In my opinion, the sentence should be grammatically rearranged since it seems to me that in the current form, there is no predicate in there.
Line 46: „100 million mu” – Authors should clarify what the „mu” means
Line 109: this sentence „Prior to delignification, prepare SC solutions with varying mass fractions” should be rearranged for better quality.
In the materials and methods: somewhere in this section should be paragraph devoted to the strains used, especially since these strains are crucial for efficient C5 and C6 fermentation from the hydrolysed poplar
Line 112: „Suction filtration” means vacuum-assisted filtration?
Line 114: appropriate citation according to NREL methodology should be added here
Lines 116-117: „Glucose, xylose, and 116 arabinose were analyzed using a two-step acid solution.” – this method is not widely known, the sentence is not clear to me and it should be rearranged with the proper methodology description or citation.
Lines 120-124: This sentence should be rearranged for clarity, maybe divided into to or three shorter ones. Additionally – „triangular flask” seems strange to me, I suggest „Erlenmayer’s flask” or the „conical flask” description instead (if the triangular flask is the same)
Lines 126-130: this part should be rearranged from the grammatical point of view – The starting sentence in the lines 126-127 looks like was not properly finished. In the line 130 there is probably the unwanted corruption „acidg”.
Line 134: „Where CHPLC is the total concentration of glucose, xylose and arabinose in the liquid 134 phase assay” – I’d consider to change it into: „Where CHPLC is the total concentration of glucose, xylose and arabinose in the hydrolyzate” for better clarity.
Line 136: „ContentSuger” – was it intentional or it should be „ContentSugar” ? - there are a few similar words „Suger” in the rest of the manuscript – if it is a misspelled „sugar”- it should be changed e.g Line 157; 160; 272?
Lines 138-140: What was the flask volume? In my opinion, the clarified and more defined option for expressing the yeast concentration is „g/L” unit. Is it possibile to change OD into g/L ?
Table 2 , Table 3 – „solid yield” expression -in my opinion, it should be defined somewhere in the text, since it is not obvious what it means in this manuscript.
Line 234: „ligninized poplar” or „deligninized poplar” ? maybe „delignified poplar” ?
Lines 213-232 – „Instead…..value” – this sentence should be rearranged since it seems like it should be some verb/activity description/ (predicate, with the proper grammar form) in it.
Line 236: „on a solid load delignification raw material” – it should be clarified (grammar)
Line 283: „At an H2SO4” in my opinion it should be: „At the H2SO4”
Figure 3: The unit values at these graphs are too small to read when it is A4 sheet printed- the units fonts (on X,Y,Z axes) should be enlarged
Figure 5 the titles of X, and - especially Y axis, should be smaller -> then it will be possible to place the whole Y axis’s title in one line?
Figure 7 – In my opinion, these strains should be defined/described in the materials and methods.
Comments on the Quality of English Language
I’m not a native English user but I think that the authors should carefully check the whole text for grammar mistakes since in the current form there are many (as I enlisted above) grammar imperfections affecting the text clarity.
Reviewer 2 Report
Comments and Suggestions for Authors
The article entitled “Highly Efficient Production of Cellulosic Ethanol From Poplar Using an Optimal C6/C5 Co-Fermentation Strain of Saccharomyces Cerevisiae, presents very interesting results, however some details need to be adjusted.
Introduction
L: 32, Please exchange for more recent data on biofuel production.
I think that in the introduction the following information should be added, Quantity of bioethanol produced in the world and quantity of second generation ethanol currently being produced.
Materials and Methods
I think the materials and methods are suitable
Results and Discussion
L: 167-177 a deeper analysis of the physical-chemical composition of biomass must be carried out. More examples of other types of biomass should be brought. Furthermore, compare with other works that had used this type of biomass.
Please improve the resolution of the figures
L: 293. The Box-Behnken design was used to optimize the response value of hydrolysis experiment of delignification feedstock.
I think that rather than using acid concentration (% w/v) as a variable, the independent variable really is pH because it directly influences enzyme activity. This is because the system may be highly buffered and a small variation in the acid concentration may not change the pH of the system. So I ask to change this variable by pH. Column 2 of table 3. So, from then on, carry out the analysis using pH as a variable and not (% w/v).
I think that in this work we need to do a total mass balance of the system at the end. From pretreatment to ethanol production.
Add error bars to all figures
Please rewrite the conclusion
References are adequate
Round 2
Reviewer 2 Report
Comments and Suggestions for Authors
I think the new version of the paper entitled “Highly Efficient Production of Cellulosic Ethanol from Poplar Using an Optimal C6/C5 Co-Fermentation Strain of Saccharomyces Cerevisiae” has been significantly improved. However, I think that minor adjustments are still needed.
For example,
The resolution of Figure 1 still needs to be improved.
I did not observe any changes regarding this question “I think that rather than using acid concentration (% w/v) as a variable, the independent variable really is pH because it directly influences enzyme activity. This is because the system may be highly buffered and a small variation in the acid concentration may not change the pH of the system. So I ask to change this variable by pH. Column 2 of table 3. So, from then on, carry out the analysis using pH as a variable and not (% w/v) ”
Please add a column in Table 3 with the pH of each experimental unit.
Thank you very Much
